# A Systematic Review and Meta-Analysis of Virtual and Traditional Physical Activity Programs: Effects on Physical, Health, and Cognitive Outcomes

**DOI:** 10.3390/healthcare13070711

**Published:** 2025-03-24

**Authors:** Đorđe Hajder, Bojan Bjelica, Saša Bubanj, Nikola Aksović, Milan Marković, Radenko Arsenijević, Gabriel-Stănică Lupu, Tomislav Gašić, Constantin Sufaru, Lazar Toskić, Daniel-Lucian Dobreci, Tatiana Dobrescu, Mihai Adrian Sava

**Affiliations:** 1Faculty of Physical Education and Sports, University of East Sarajevo, 71126 Lukavica, Bosnia and Herzegovina; hajderdjordje@yahoo.com (Đ.H.); vipbjelica@gmail.com (B.B.); 2Faculty of Sport and Physical Education, University of Niš, 18000 Niš, Serbia; 3Faculty of Sport and Physical Education, University of Priština-Kosovska Mitrovica, 38218 Leposavić, Serbia; kokir87np@gmail.com (N.A.); milan.markovic@pr.ac.rs (M.M.); radenko.arsenijevic@pr.ac.rs (R.A.); lazar.toskic@pr.ac.rs (L.T.); 4Faculty of Movement, Sports and Health Sciences, Vasile Alecsandri University, 600115 Bacau, Romania; gabi.lupu@ub.ro (G.-S.L.); sufaruconstantin@ub.ro (C.S.); dobreci.lucian@ub.ro (D.-L.D.); sava.adrian@ub.ro (M.A.S.); 5High School Center Prijedor, Republika Srpska, 79000 Prijedor, Bosnia and Herzegovina; gasictomislav@yahoo.com

**Keywords:** virtual reality fitness, traditional exercise, digital physical activity, cognitive performance, psychological well-being, review

## Abstract

(1) **Background**: The aim of this study was to conduct a systematic literature review and meta-analysis on the effects of virtual reality (VR) and traditional (TR) physical activity programs, analyzing their impact on the physical, health, and cognitive aspects of participants. The study sought to identify the advantages and limitations of both methods, taking into account previous research and potential areas for future studies. (2) **Methods**: The study protocol for this systematic review was registered at the International Platform of Registered Systematic Review and Meta-analysis Protocols (INPLASY202530015). The review followed PRISMA guidelines, and studies were selected based on their relevance to the research objectives using the PICOS model criteria. The authors applied a meta-analysis in addition to a systematic review to further ensure the accuracy of the results. Primary outcomes included physical and cognitive performance, while secondary outcomes encompassed participant perceptions and psychological effects. (3) **Results**: The findings indicate that VR training significantly enhances flexibility, motivation, and cognitive abilities, particularly in populations with limited access to traditional exercise methods. The most pronounced effects were observed in programs lasting 8 to 12 weeks with a moderate to high intensity. In contrast, TR showed superiority in developing strength, endurance, and cardiorespiratory functions. (4) **Conclusions**: VR offers significant benefits as an adjunct or alternative to TR, especially for individuals with limited resources or physical accessibility. However, variations in methodological approaches, short program durations, and sample heterogeneity highlight the need for further longitudinal research. Standardizing VR training duration and intensity is essential to ensure consistent and reliable outcomes.

## 1. Introduction

According to the World Health Organization (WHO), due to urbanization, a lack of recreational opportunities, and limited free time, 80% of adolescents and 25% of adults worldwide are physically inactive [1]. Physical inactivity contributes to an increased risk of diseases such as type 2 diabetes, colon cancer, and coronary heart disease, accounting for approximately 9% of global mortality [1,2]. The integration of virtual reality (VR) technology presents a new approach to physical activity and health promotion [3], offering an environment in which external factors such as lighting and weather conditions do not play a crucial role, as they do in traditional exercise [4].

Recent studies, particularly those conducted during the COVID-19 pandemic, raise the question of whether VR can serve as a substitute for traditional training and what its benefits are. Over the past decade, VR-based training has gained prominence, ranking among the most modern fitness solutions. It has enabled older adults [5], individuals from rural areas, and those with limited free time to engage in adequate physical activity through VR emulators [6]. The growing relevance of VR is evident, as studies have shown that participants in VR-based exercise programs achieve significantly better results in balance, coordination, strength, and flexibility compared to those engaging in traditional exercise.

VR creates a digital training environment without physical boundaries, where the system itself monitors the user’s activity. The ability to personalize training programs allows for its application across all levels of experience, from beginners to professional athletes. Initially, VR systems required players to use a mouse to control movement and speed [7,8]. However, technological advancements have led to the development of integrated sensors synchronized with computers, forming head-mounted displays (HMDs) and handheld joysticks that enable full immersion in virtual environments [9]. Today, platforms such as PlayStation VR, Oculus Rift, HTC Vive, and VrFit, along with Xbox 360 and Nintendo paired with motion sensor cameras, represent cutting-edge technologies for tracking and controlling avatar movements during exercise [10]. VR-based activities range from gravity-free ball catching [11] and virtual cycling [12,13] to interactive training with virtual coaches [14] and arcade-style games requiring physical movement to hit targets [15,16].

Traditional forms of exercise, such as running, cycling, or strength training, have long been recognized for their benefits to cardiovascular health, muscle mass development, fat reduction, and mental well-being [17]. However, accessibility to traditional exercise is often limited by time, space, and equipment availability. In contrast, VR-based exercise offers flexibility and accessibility, allowing users to train at home and tailor activities to their needs [18]. Research indicates that VR training can enhance motor skills, increase motivation due to its interactive and engaging nature, and serve as an effective alternative for individuals with restricted access to traditional training methods [19].

While VR exercise is not a full replacement for traditional physical activity, Zeng and associates demonstrated that it serves as a highly effective alternative, particularly for individuals with limited access to conventional exercise options [20]. VR-based training has been shown to reduce pain perception, anxiety, and fatigue while aiding in the rehabilitation of upper and lower limbs, stroke recovery, and post-traumatic stress disorder treatment [21,22]. Additionally, 2D-based VR applications have been utilized to promote general physical activity [23,24] and enhance relaxation by reducing stress and improving cognitive function, especially in response to the imbalances caused by the COVID-19 pandemic [25,26,27]. The positive effects of VR training extend to motor skills, physical activity levels, and perceptual-cognitive abilities, contributing to improvements in several biomechanical measures [28,29,30]. VR has become increasingly relevant in physical activity due to its accessibility, interactivity, and the elimination of time and space constraints. It offers a high level of motivation and adaptability, contributing to its growing application in fitness and sports training.

While the benefits of VR in physical activity have been extensively studied, its limitations and challenges—such as the standardization of methods, individual adaptability, and long-term effects on motor abilities—remain insufficiently explored. A systematic review and meta-analysis of existing evidence are necessary to identify research gaps and establish directions for future studies. Additionally, despite its advantages, VR training faces challenges that may affect its effectiveness and applicability. One of the main concerns is long-term adherence, as initial user enthusiasm may decline over time [14]. Moreover, certain populations experience issues such as motion sickness and disorientation, limiting accessibility [17]. The effects of VR training also vary across different demographic groups depending on age, fitness level, and prior experience with digital technologies [31]. Therefore, a deeper understanding of how VR can be optimized for different users is crucial. This systematic review and meta-analysis not only examines the benefits of VR training but also explores key challenges and limitations, particularly factors influencing outcome variability. Furthermore, the study aims to identify aspects such as the optimal duration of VR programs and the potential for combining traditional and VR-based methods to maximize their effectiveness.

Therefore, the aim of this study was to conduct a systematic literature review and meta-analysis on the effects of VR and traditional (TR) physical activity programs, analyzing their impact on the physical, health, and cognitive aspects of participants. The study sought to identify the advantages and limitations of both methods, considering previous research and potential areas for future studies.

To achieve this objective, the following tasks were undertaken:A systematic search of electronic databases;A review of the collected literature.

## 2. Materials and Methods

A systematic and transparent approach to data collection, analysis, and selection is essential for ensuring validity, minimizing bias, and maintaining the reliability of findings. For this reason, the PRISMA guidelines (Preferred Reporting Items for Systematic Reviews and Meta-Analyses) were applied in this study [32]. The study protocol for this systematic review was registered with the International Platform of Registered Systematic Review and Meta-analysis Protocols (INPLASY202530015).

### 2.1. Search Strategy

The electronic search was conducted across multiple databases, including PubMed, Web of Science, Scopus, MEDLINE, ERIC, and Google Scholar, as well as other relevant literature that could provide insights into the research problem. The analysis focused on studies published between 2008 and 2024. A combination of keywords related to virtual reality and physical activity was used to refine the search strategy. This study included 23 research articles that met strict methodological criteria and applied relevant interventions.

The search strategy included the following keywords: “virtual reality” AND “physical activity” OR “exercise” NOT “gaming”, with Boolean operators applied to refine the results. The specific keywords used in the database searches were as follows: Effectiveness, Virtual Training, Digital Exercise, Online Fitness, and Influence. Conference abstracts, papers, preprints, and other forms of gray literature were excluded from this review to ensure that only peer-reviewed and validated sources were analyzed. Additionally, only studies published in Serbian and English were included, due to the research team’s knowledge of these languages, which ensured accurate data extraction and interpretation (Table 1). While this limitation may restrict access to relevant studies in other languages, automated translation tools were considered insufficient for handling complex methodological analysis. This decision was made to ensure the validity and reliability of the results.

Titles and abstracts were initially screened to determine relevance, and only studies that met the predefined inclusion criteria were selected for further analysis. To enhance the sensitivity of this systematic review, the search strategy was tailored for each database when possible. If essential data were missing from the included studies, the authors contacted the study authors for clarification. However, if non-critical data were missing, they were excluded from Table 2 and not discussed to avoid potential misinterpretation of the results.

### 2.2. Inclusion and Exclusion Criteria

To ensure an objective selection process, three authors independently assessed studies based on predefined inclusion and exclusion criteria. The PICOS framework (Population, Intervention, Comparators, Outcomes, and Study Design) was used to identify eligible studies. The inclusion and exclusion criteria, along with the results of the descriptive statistical analysis, are summarized in Table 1.

### 2.3. Study Selection and Methodological Quality

Data screening and extraction were conducted by three authors (B.B., R.A., and L.T.) using the previously described search protocol. EndNote software (v. 21) was utilized for citation management, while Mendeley reference management software (v. 2.111.0, Copyright © 2024 Elsevier Ltd., Barcelona, Spain) was used for duplicate detection. The study critically evaluated the quality of the included research and systematically identified potential limitations, such as sample variability and inconsistencies in VR instrumentation, which are discussed in detail in the discussion section.

The methodological quality of the selected studies was independently assessed using the Physiotherapy Evidence Database (PEDro) scale, which comprises 11 criteria (Table 3). Each criterion was evaluated with a binary scoring system (+ or −) or an alternative numerical representation (1 or 0). Studies scoring six or more points were categorized as high quality, those scoring between 4 and 5 were categorized as moderate quality, and those scoring below 4 were categorized as low quality. To further improve the study’s methodology and assess the quality of the studies, the authors also applied the Cochrane risk of bias tool (Table 4). The quantitative score was assigned as follows: Low risk of bias = 0; Uncertain risk of bias = 1; and High risk of bias = 2.

The overall quality score for each study was calculated by summing the individual item values using the following scale: overall score ≤ 1 (high quality), overall score ≤ 3 (moderate quality), overall score > 3 (low quality). In cases in which essential data were missing from selected studies, the corresponding authors were contacted for clarification. If the missing data were not critical to the primary analysis, the study was retained, and such data were excluded from specific comparisons. However, studies with incomplete key variables were excluded to ensure data integrity and comparability.

Furthermore, to maintain methodological rigor, gray literature (e.g., conference abstracts, dissertations, and preprints) was excluded. This decision was based on concerns regarding the lack of peer review, potential methodological inconsistencies, and difficulties in verifying the accuracy of reported data. By focusing exclusively on peer-reviewed sources, we aimed to enhance the reliability and reproducibility of our findings. Disagreements were resolved through discussion between two researchers, with a third researcher consulted if a consensus could not be reached. The selection of relevant studies was carried out independently by two authors (N.A. and S.B.), who cross-verified their choices for accuracy. A third author (R.A.) made the final decision on study inclusion. The authors applied a meta-analysis in addition to a systematic review to further ensure the accuracy of the results (Table 5). The Fisher test was used to analyze the significance of differences between experimental groups in the final measurement (Table 6).

## 3. Results

To enhance the clarity and consistency of the findings, numerical values have been provided wherever possible. However, some outcomes were reported qualitatively due to variations in study methodologies and measurement tools. While reductions in BMI are explicitly quantified, other parameters, such as motivation and cognitive improvements, are often described descriptively in the reviewed studies. Future research should aim to standardize reporting metrics and ensure that both primary and secondary outcomes are quantified whenever feasible to facilitate direct comparisons.

### 3.1. Literature Characteristics

The process of data collection, analysis, and study elimination is illustrated in Figure 1. An initial search across the selected databases identified 378 potential studies. After removing duplicates, screening abstracts and titles, and applying other exclusion criteria, 60 papers remained. Following a more detailed eligibility assessment, 38 papers were excluded, leaving 22 studies that met the predefined inclusion criteria and were included in the systematic review. The full procedure for study selection and analysis is presented in Figure 1, the distribution of research by databases is presented in Figure 2, the comparison of the effectiveness of VR and TR across different categories is presented in Figure 3, and the meta-analysis forest plot figure, with the effects of VR-based interventions across various domains in Figure 4.

### 3.2. Characteristics of the Studies

Six studies focused exclusively on male participants [40,43,44,45,46,47], while five studies involved only female participants [14,15,33,34,41]. Ten studies included both male and female participants [16,17,30,31,35,36,37,39,42,48]. Proffitt and colleagues [38] did not specify the gender of the participants.

Age Groups: The studies were categorized into two age groups: younger populations aged 18 to 28 years [37,40,43,44,46,47] and adults aged 50 and over [11,15,20,31,34,38,41].

Different VR and TR Programs Used: All studies utilized VR programs that were focused on motor skills, body composition, or cognitive functions [20,34,35,37,41,43,44,45,46]. Most studies compared VR with traditional training methods, such as aerobic exercise, strength training, and flexibility [14,31,38,42,47,48].

Research Outcomes—Physical, Health, and Cognitive Changes: Most studies measured physical outcomes, such as body composition, flexibility, and cardiorespiratory function [31,34,37,41,42,44]. Seven studies reported reductions in depression and anxiety, as well as increases in motivation [14,15,31,38,40,43,46]. Six studies confirmed improvements in attention, concentration, and mental agility [17,31,37,38,43,45].

Program Duration: Program durations ranged from 6 to 26 weeks, with moderate to high training intensity. Programs lasting 8 to 12 weeks yielded the most optimal results [31,40,43,46]. Activity intensity ranged from 50% to 75% of the participants’ maximal effort [34,37,41].

### 3.3. The Methodological Quality Assessment of the Included Studies

According to the PEDro scale, fifteen studies (*n* = 15) were classified as high quality, seven studies (*n* = 7) were classified as moderate quality, and no low-quality studies were included in this systematic review (Table 3). The literature review was conducted following PRISMA guidelines. Both longitudinal and cross-sectional studies were included in the analysis, focusing on the effects of VR on physical and cognitive abilities.

According to the Cochrane risk of bias tool, sixteen studies (*n* = 16) were classified as high quality, six studies (*n* = 6) were classified as moderate quality, and no low-quality studies were included in this systematic review (Table 4).

## 4. Discussion

The primary results indicate that VR significantly contributes to improvements in participants’ psychological well-being, with positive effects also observed on cognitive functions. However, traditional methods appear to be more effective in developing motor fitness abilities. Secondary findings suggest a positive impact of VR on participants’ satisfaction, particularly in populations with limited access to fitness centers. The included studies exhibited high variability in respondent samples, the use of different VR devices, and the outcomes themselves, which represents a limitation of this study. The research underscores the need for further analyses involving longer time periods and more clearly defined participant populations to better explore the potential of VR and promote its more frequent use. This systematic review and meta-analysis included studies with diverse VR interventions, populations, and study designs, yet a formal subgroup analysis was not conducted. Differences between interactive gaming-based VR programs and guided VR workouts, as well as variations in training duration and intensity, remain areas for further investigation. Future research should conduct stratified analyses to determine whether specific VR modalities yield superior effects for particular populations or outcomes.

### 4.1. Comparative Analysis of Traditional and Virtual Training Methods

VR and traditional training methods represent two fundamentally different approaches to physical activity, each with its own advantages and limitations. Analyzing the available research reveals that both approaches can be effective in different aspects of physical abilities. While the effects on health parameters are significant, it is important to note that the two methods produce distinct outcomes.

VR often relies on interactive technology and simulations that allow users to train in “realistic” conditions and, due to different consoles, in controlled environments. Seo and associates [14] indicate that VR training can significantly reduce depression and increase motivation for exercise, while also having positive effects on basic health parameters such as body mass index. In contrast, traditional approaches typically involve trainers and established exercise protocols, in which direct contact with the trainer is crucial for movement correction and motivation.

However, research critiquing VR methods should also be considered. Nitkiewicz and associates [17] note that excessive reliance on VR systems can lead to reduced attention and continuity in training, particularly in individuals with lower levels of physical fitness. Traditional methods, on the other hand, require fewer resources and are more accessible to the general population.

McClure and associates [48] suggest that VR training is as effective as traditional methods. Rubio-Arias and colleagues [49] argue that both methods provide a moderate intensity, but the traditional method is viewed less favorably due to reduced motivation to stick with the exercise program. Participants themselves report perceiving traditional exercise as less enjoyable. Interestingly, VR outperformed traditional training in terms of improvements in strength, balance, and flexibility. After a 5-week experimental treatment, VR significantly influenced these parameters, while the traditional method did not show statistically significant improvements. These findings are supported by earlier research [50].

The findings of this review are consistent with other studies on the same topic [51,52]. There is little doubt that VR training programs are effective in increasing physical activity frequency, enhancing physical performance, and inducing positive mental changes.

While many studies have demonstrated positive effects of VR training, some findings highlight certain limitations and contradictions. For instance, while some authors have reported significant improvements in flexibility, motivation, and cognitive abilities following VR interventions [14,31], others have found less pronounced effects or no significant changes, particularly in strength and endurance [17,48]. Additionally, some studies suggest that VR training may lead to reduced attention and training continuity in certain users, which can affect long-term benefits [17]. These discrepancies indicate the need for further research to clarify under which conditions and for which populations VR training yields the most reliable results.

Regarding the duration of VR training, previous studies suggest that programs lasting 8 to 12 weeks are the most effective for enhancing cognitive and motivational aspects [31,46]. However, a systematic comparison of different intervention lengths is lacking, making it premature to claim this period as optimal. For example, while some studies have shown significant effects after six-week programs [40,47], others have reported improvements only after 12 weeks or longer [43]. Future research should systematically analyze the duration and intensity of VR training to determine the best combination for different populations.

In terms of practical implications, VR training has been particularly effective among older adults [31], individuals with limited access to traditional training [14], and beginners seeking interactive and engaging workout methods [15]. However, in athletic populations requiring high levels of physical exertion, VR training may be less effective due to the lack of physical resistance in digital exercises [48]. These findings suggest that VR programs must be tailored to the specific needs of users to achieve optimal outcomes.

A key strength of this research is the inclusion of high-quality studies that have been rigorously analyzed, ensuring the validity of the results. The systematic review and meta-analysis include both longitudinal and cross-sectional studies, which highlight the effects of VR and its advantages and disadvantages when compared to conventional methods.

### 4.2. Impact of Virtual Training Programs on Physical, Health, and Cognitive Aspects

The findings of this meta-analysis (Table 5; Figure 4) provide valuable insights into the effects of virtual reality (VR)-based interventions across various domains, as demonstrated by the studies included in the final forest plot and summary table. The calculated effect sizes (Cohen’s d) consistently indicate a positive impact of VR interventions, with values ranging from 0.38 to 0.75, suggesting moderate to strong effects depending on the specific study.

VR training methods and their impact on health and motor skills have become an increasingly important and frequently discussed topic. A review of research has demonstrated the positive effects of virtual exercise on motor skills and health parameters.

The results presented in Table 6 indicate a high level of statistical significance across all analyzed aspects, with cognitive factors showing the most pronounced effect (*p* = 0.0022). The data were obtained based on the *p*-values of the experimental groups, and an overview of the analyzed variables is provided in Table 2. The studies that were not included in the statistical analysis are [17,39], because they did not report *p*-values, and these were most often studies based on questionnaires and self-reports. Some studies did not have *p*-values for certain variables, while they were available for others [14,16,35].

Among the included studies, Wang [31] reported an effect size of 0.72 (95% CI: 0.50–0.94), indicating a substantial benefit of VR interventions in functional fitness improvements. Similarly, Irving and associates [41] presented the highest effect size (0.75, 95% CI: 0.55–0.95), reinforcing the effectiveness of VR-based training for skill acquisition and rehabilitation. In contrast, McDaniel and associates [37] reported the lowest effect size (0.38, 95% CI: 0.20–0.55), highlighting potential variability in intervention effectiveness across different populations and methodologies. The study by de Melo Ghisi and associates [35] was not included in the meta-analysis because the effect size was not calculated; instead, the data from this study were analyzed qualitatively. The lack of necessary data for effect size calculation may have been due to the reporting method, the statistical approaches used, or the absence of relevant quantitative data in the study. Without this information, it was not possible to standardize the results and include them in further analyses. It is important to emphasize that, although only one study was analyzed qualitatively, this will not compromise the validity of the meta-analysis, as the reasons for its exclusion have been transparently explained.

It is important to note that in most studies with a four-week intervention, the results were statistically insignificant, whereas the best effects were achieved in programs lasting a minimum of eight weeks or more. This finding further emphasizes the importance of longer interventions in achieving sustainable and measurable improvements in the physical, cognitive, and health aspects of participants.

Studies focusing on data collection through surveys and the subjective self-perceptions of participants were excluded from this statistical analysis.

All the studies cited in this paper show positive changes in the body composition, motor skills, cardiovascular system, and cognitive characteristics of the participants.

For example, the application of VR fitness programs has led to reductions in BMI [14,33,34,40,41,44], body fat percentage [37,41], and waist and hip circumference [33,41,44]. Research by various authors [31,34,42] indicates that the most significant improvements are seen in coordination and balance. Additionally, studies [30,31,40] show that VR programs, including stretching exercises, can significantly increase the range of motion, reduce the risk of injury, and benefit the maintenance of functional body abilities, such as joint mobility, while positively impacting daily activities. Research on explosive training [46], repetitive static exercises [44], and dynamic strength training [30] using VR [17,31,36,40,44] demonstrates significant improvements in muscle strength and endurance. The exercises were adapted to various fitness levels to achieve equivalent results and develop functional abilities [37], thereby increasing strength and endurance.

A key benefit of VR training is its positive impact on cognitive characteristics, such as reducing depression and anxiety [39]. Engaging in physical activity through VR platforms leads to an increase in endorphins—the “happiness” hormone—which plays a crucial role in improving mood, thereby reducing symptoms of depression and anxiety [14,39,40,46]. Participants in these studies reported emotional improvements, with reductions in negative emotions like sadness and depression. VR training also reduces tension, anxiety, and stress levels. Various types of physical activity within VR moments stimulate the brain, promoting mental sharpness and agility [36]. Significant effects of VR on maintaining attention, as well as enhancing concentration, motivation, and focus during activities, have also been observed [17,38,39,47].

VR training has also been found to positively impact blood pressure regulation, including both diastolic and systolic readings, as well as pulse rate [33,41]. VR workouts, such as brisk walking, running, or cycling, have special effects on regulating blood pressure, which can reduce the risk of heart disease and stroke. Research shows that VR training positively affects cardiorespiratory [31] and cardiovascular endurance [34,48], reducing participants’ resting heart rate. Studies indicate a decrease in heart rate during daily activities, lowering the risk of heart disease [34,48]. Wang [31] confirms that VR positively affects the cardiorespiratory system by improving lung and heart function.

In addition to improvements in motor skills, body composition, cognitive characteristics, and the cardiovascular system, VR also positively affects personal impressions and later engagement in activities. Research by [15,16,38,43,45] suggests that VR technology can transform training by making it more enjoyable and dynamic. These features of VR training allow users to stay focused on their training and exercises because they perceive them as realistic and exciting. As a result, participants remain motivated and engaged for longer periods, leading to better outcomes.

Studies on social interaction in VR [14,15,16,35,39,43,45,46] also demonstrate that VR has a positive impact on users’ social connections. It creates opportunities for social interaction, allowing participants to exchange experiences, provide motivation, and offer support. Additionally, VR helps reduce the increasing sense of loneliness and its negative effects on emotional and psychological health.

Although the research mentioned above clearly confirms the broad applications and benefits of VR in physical activity, further studies continue to explore the potential applications of VR.

### 4.3. Limitations of the Study

The systematic review and meta-analysis aimed to provide an overview of VR and TR methods and highlight the differences between them. However, several limitations should be considered when interpreting the results.

First, the included studies involved participants from various age groups with various health conditions and fitness levels. This diversity makes it challenging to generalize the results to a broader population and reduces the consistency of the conclusions. Second, the wide range of VR devices and training methods used further complicates the comparison of results. Different technologies offer varying levels of interaction and engagement, which can significantly influence study outcomes.

Another limitation of this study is the relatively small sample size of 23 selected research articles, which may restrict the comprehensiveness and generalizability of the findings. While a larger sample could have broadened the scope, the focus was on including high-quality, methodologically sound studies to ensure the reliability of the results, prioritizing quality over quantity.

A potential limitation of VR-based training is the lack of physical resistance in virtual environments, which may affect strength development compared to traditional resistance training methods. Additionally, some users may experience motion sickness or reduced adherence over time as the novelty diminishes. These factors should be considered when integrating VR into long-term fitness programs, and future studies should assess their impact on training outcomes.

Finally, another limitation is the duration of the included programs. Most studies lasted only a few weeks, which is insufficient to assess the long-term effects of VR training on fitness and health parameters, as well as to make meaningful comparisons with TR methods. Many studies also relied on participants’ subjective self-reports of their physical capabilities and lifestyle habits, which introduces potential bias and limits the objectivity of the findings.

## 5. Conclusions

VR physical activity programs have demonstrated significant benefits in improving flexibility, motivation, and psychological characteristics. In contrast, traditional exercise (TR) remains essential for developing strength, endurance, and cardiorespiratory fitness. These results suggest that both approaches complement each other in modern physical activity programs.

The optimal results of VR training were observed in experimental treatments lasting between 8 and 12 weeks with a moderate to high intensity. This duration allows for a sufficient amount of time for adaptation and the achievement of notable physical and psychological changes, while the intensity level ensures adequate stimulation of motor and cognitive abilities. The claim that an 8–12-week duration is optimal for VR training is based on a subset of studies reporting significant improvements within this timeframe. However, due to the variability in study designs and intervention structures, a systematic comparison of different training durations is necessary to confirm whether this period is truly the most effective. Future research should explore the impact of varying durations, frequencies, and intensities of VR training to establish evidence-based recommendations.

While the findings of this study indicate that VR training has notable benefits, particularly in enhancing motivation, cognitive engagement, and accessibility, the high variability across studies suggests that its effectiveness is context dependent. The superiority of VR over traditional training cannot be definitively concluded without further high-quality research that includes standardized outcome measures and statistical comparisons.

Although both methods offer distinct advantages and drawbacks, the limitations of this study—such as the sample heterogeneity, VR device variability, and short study duration—highlight the need for further research to establish clear guidelines for implementing VR training across different populations. Additionally, standardizing protocols, including consistent measures of intensity and duration, is essential for enhancing the reliability and practical application of these methods. These findings further confirm the significance of VR technology as an innovative approach to physical activity, particularly for populations with limited access to traditional exercise methods. By combining physical, psychological, and cognitive benefits, VR training can contribute to improving overall health and quality of life. Future research should focus on the long-term effects of VR training and its potential integration into various health and sports programs.

## Figures and Tables

**Figure 1 healthcare-13-00711-f001:**
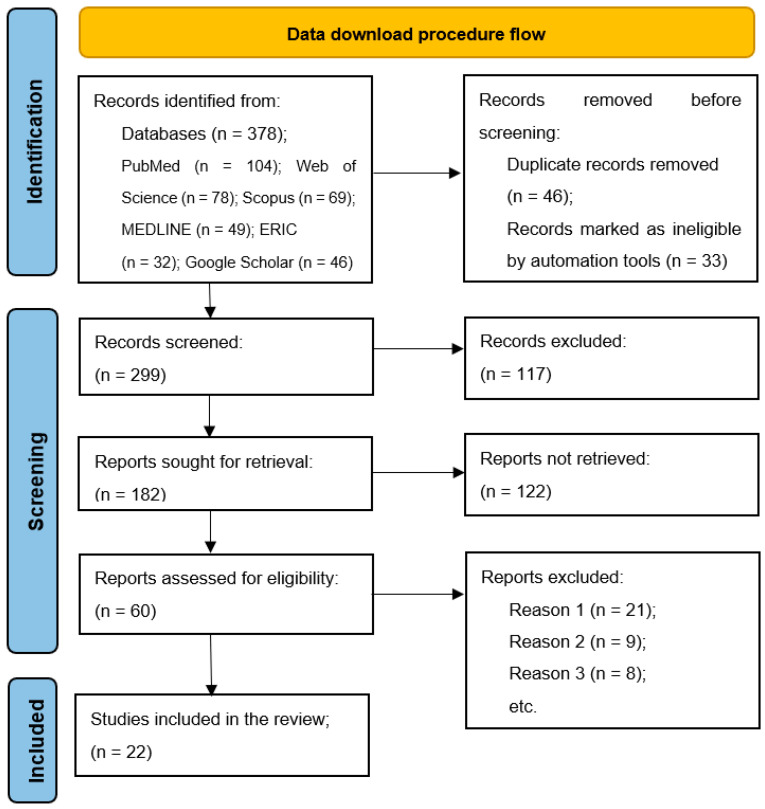
Flow chart diagram of the study selection.

**Figure 2 healthcare-13-00711-f002:**
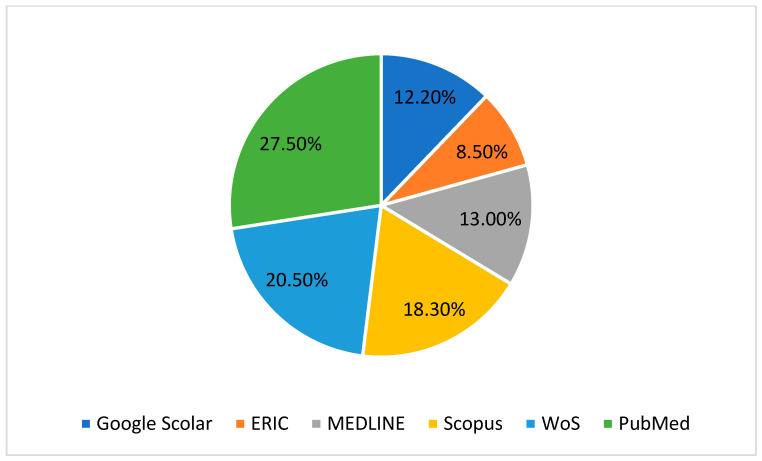
Distribution of research by databases.

**Figure 3 healthcare-13-00711-f003:**
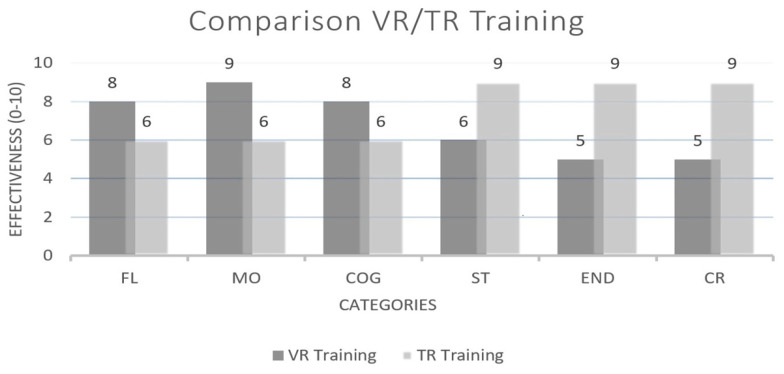
Comparison of the effectiveness of virtual (VR) and traditional (TR) training across different categories. Legend: FL—Flexibility, MO—Motivation, COG—Cognitive, ST—Strength, END—Endurance, and CR—Cardiorespiratory.

**Figure 4 healthcare-13-00711-f004:**
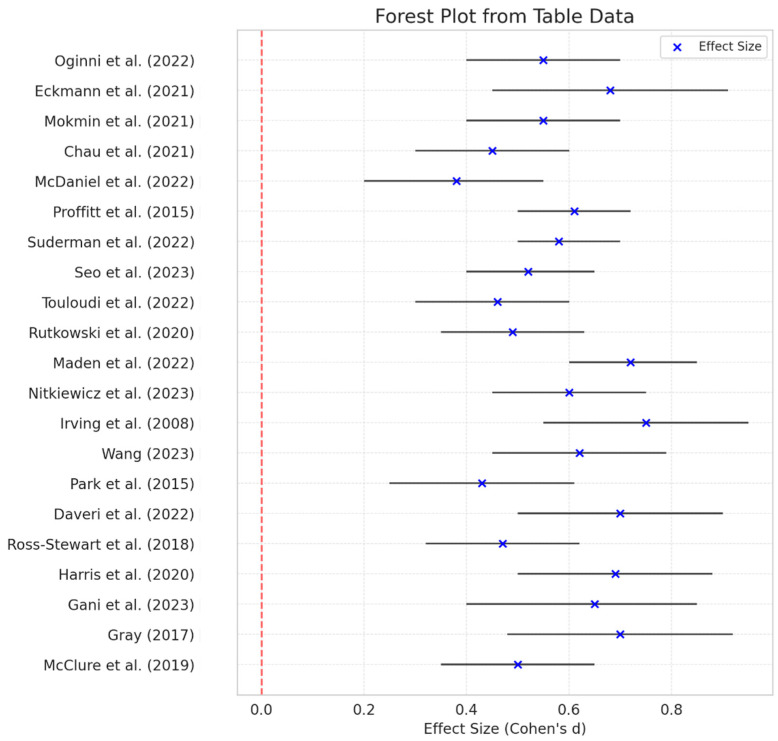
Meta-analysis forest plot figure, with the effects of virtual-reality-based interventions across various domains [14,15,16,17,30,31,33,34,36,37,38,39,40,41,42,43,44,45,46,47,48].

**Table 1 healthcare-13-00711-t001:** The inclusion and exclusion criteria used according to the PICOS model.

PICOS Category	Inclusion Criteria	Exclusion Criteria
P (Population)	Men and women, regardless of restrictions on lifestyle, age, and health status.	People with serious health problems or injuries that prevent physical activity.
I (Intervention)	Papers in which an experimental procedure was used in which several groups participated, as well as papers in which data were collected through questionnaires or interviews.	Studies with incompatible virtual methods, studies whose primary focus was not physical and cognitive training but rather participants’ impressions, and studies that relied solely on self-reported physical activity data without objective verification were excluded.
C (Comparators)	Studies comparing experimental and control groups (e.g., VA vs. CG, IPE vs. NA, etc.) or comparing different groups within virtual training (e.g., EXP1 vs. EXP2, PA vs. VA).	Studies between unrelated sports (e.g., men’s basketball players vs. handball players or soccer players).
O (Outcomes)	Effects of virtual activity (VA-VR) on physical and cognitive abilities after implementation of the program; Information about the personal impressions of participants.	Incomplete results.
S (Study design)	Randomized and non-randomized controlled studies; Longitudinal and transferal studies; Studies written in Serbian and English.	Duplicates; Conference papers and abstracts; Case reports (e.g., <5 participants per group); Review articles; Preprints; Inappropriate frame of analysis in the period between 2008 and 2024; Studies written in a language that was not Serbian or English.

**Table 2 healthcare-13-00711-t002:** A systematic review of the included studies.

References	Sample	Testing/Instrument	Exposure	Dose	Findings
Oginni et al., 2024 [33]	n = 30 ♀ y = 37.8 ± 8.8	Blood pressure, Waist, Hip, Heart rate, Circumferences, Physical activity readiness questionnaire, Body mass index	VA-VCZ PA-SPA CG-NA	2 × 60 min 6 w	Diastolic blood pressure ↓ Body mass index ↓ Waist, hip ↓
Eckmann et al., 2021 [34]	n = /♀ y = 45–65	Body mass index, Bioelectrical impedance, Heart rate, Motor ability	VA-WBP Well-WBP	3 × 60 min 10 w	Body mass index ↓ Motor ability ↑ Heart rate ↓
de Melo Ghisi et al., 2024 [35]	n = 160 ♀ ♂ y = /	Survey, Mediterranean food, Pedometer	VA IPE	- 12–24 w	Mediterranean food ↑ Pedometer ↑ Survey ↑
Mokmin et al., 2021 [16]	n = 33 ♀ n = 21 ♂ y = 19–24	Survey, Interview	VA-BMTMCTOM	26 w	Survey ↑ Interview ↑
Chau et al., 2021 [36]	n = 135 ♀ ♂ y = 62.7	Upper extremities, Lower extremities, Cognitive functions	VA PA-TUG, BBT CG-HK-MoCA 5-min, BTO	3 × 30 min 6 w	Upper extremities ↑ Lower extremities× Cognitive functions ↑
McDaniel et al., 2022 [37]	n = 2 ♀ n = 11 ♂ y = 29.8 ± 6.2	Height, Body mass, Body composition, Army combat fitness test, Mean body fat percentage	VA-TRX Elite ACFT Kit PA-DC-430 U, ACFT	3 ses 12 w	Mean body fat percentage ↓ Army combat fitness test ↑ Body fat ↓ Body mass×
Proffitt et al., 2015 [38]	n = 30/ y = 50+	Health activity information form, Immersive tendencies questionnaire, Tellegen absorption scale, Minimal heart rate, Maximal heart rate, Virtual environment preferences, Perception of challenge, Focus on movement	VA-JM+MKM PA-TBT+MKM	-	Minimum heart rate × Maximal heart rate × Tellegen absorption scale ↓ Virtual environment preferences ↑ Perception of challenge ↑ Focus on movement ↑
Suderman et al., 2022 [39]	n = 127 ♀ ♂ y = 59 ± 11.4	Survey	VA-ACE PA-ACE+IN	-	Survey ↑
Seo et al., 2023 [14]	n = 75 ♀ y = 40–65	Body mass index, Depression, The amount of fun during exercise, Exercise immersion	VA-IBE (VRfit) PA-IBE	3–5 × 50 min 8 w	Body mass index ↓ Depression ↓ The amount of fun during exercise ↑ Exercise immersion ↑
Touloudi et al., 2022 [15]	n = 40 ♀ y = 20–61	Survey, Expectations, Usability or utilization, Usability or learning, Usability or pleasantness, Sense of presence or spatial presence, Sense of presence or engagement, Sense of presence or realism, Tolerability, Use in the workplace	VA-IBE (VRADA) PA-IBE	2 sess. × 15 min	Expectations ↑ Usability or utilization ↑ Usability or learning ↑ Usability or pleasantness ↑ Sense of presence or spatial presence ↑ Sense of presence or engagement× Sense of presence or realism ↑ Tolerability ↑ Use in the workplace ↑
Rutkowski et al., 2020 [30]	n = 106 ♀ ♂ y = 60 ± 5	Arm curl, Chair stand, Beck scratch, Chair sit and reach, Up and go, 6 min walk test	VA-ETPA-ET(TPR)	5× 6 w	Arm curl ↑ Beck scratch ↑ Chair sit and reach ↑ 6 min walk test ↑
Maden et al., 2022 [40]	n = 44 ♂ y = 18–28	Height, Body mass, Body mass index, Smoking status, Gaming status, The nine item Internet Gaming Disorder Scale–Short Form, The International Physical Activity Questionnaire, Senior fitness test, Maximum oxygen uptake, Beck anxiety inventory	VR (XK36) PA-ATCG	3 × 30 min 6 w	Body mass ↓ Physical activity ↑ Anxiety ↓ Repetition ↑ Flexibility ↑
Irving et al., 2008 [41]	n = 27 ♀ y = 51 ± 9	Weight, Body mass index, Fat free mass, Fat mass, Abdominal fat, Subcutaneous fat, Abdominal visceral fat, Mid-thigh fat area, Mid-thigh skeletal muscle, Waist circumference, Fasting blood glucose, High-density lipoprotein cholesterol, triglycerides, Systolic blood pressure, Diastolic blood pressure, Peak oxygen uptake, Oxygen uptake at lactate threshold, Tidal volume, Metabolic equivalent of task—heart work, Basal metabolic rate	NET LIET HIET	- 16 w	Waist circumference ↓ Systolic blood pressure ↓ Fasting blood glucose× High-density lipoprotein cholesterol× Triglycerides× Diastolic blood pressure× Subcutaneous fat ↓ Abdominal fat ↓ Abdominal visceral fat ↓ Mid-thigh fat area ↓ Body mass index ↓ Body fat× Fat free mass×Mid-thigh skeletal muscle×Fat mass×weight ↓ Peak oxygen uptake ↑ Oxygen uptake at lactate threshold× Tidal volume ↑ Metabolic equivalent of task—heart work× Basal metabolic rate×
Nitkiewicz et al., 2023 [17]	n = 4 ♀ ♂ y = 25 ± 1	Accourity, Focus	VA-OQ2+HMD PA-OT, CT	-	Occasional training ↑ 1 week break ↑ Repetition ↑ Focus ↑ Continuous training ↓ Reflexes ↓ Focus ↓
Wang, 2023 [31]	n = 78 ♀ n = 20 ♂ y = 72.16 ± 4.9	Demographic characteristics of participants, Senior fitness test	E- VA+SFT CG-SFT	1 w × 150 min 12 w	Upper body flexibility ↑ Lower body flexibility ↑ Upper body strength ↑ Lower body strength ↑ Cardiorespiratory ↑ Balance ↑
Park et al., 2015 [42]	n = 30 ♀ ♂ y = 65+	Gander, Age, Height, Weight, Sway length, Average sway speed, Timed up and go	VABE	3 × 30 min 8 w	Balance sheets ↑ Sway length ↓ Average sway speed ↑ Timed up and go ↑
Ross-Stewart et al., 2018 [43]	n = 27 ♂ y = 18–23	Skill imagery, Goals imagery, Mastery imagery, Practicing automaticity, Practicing relaxation, Practicing self-talks, Practicing imagery, Self-talk competition, Competition automaticity, Imagery competition, Negative thinking competition, Strategy imagery ability, Affect imagery, Activation, Goal setting, Automaticity, Emotional control, Imagery, Relaxation	VA-SIAQ TPSQ	2 × 15 min-1d (11 ses) 12 w	Goals imagery ↑ Skill imagery ↑ Imagery competition ↑ Practicing automaticity ↑ Practicing relaxation ↑ Practicing self-talks ↑ Practicing imagery ↑ Competition automaticity ↑ Self-talk competition ↑ Negative thinking competition ↑ Mastery imagery ↑
Daveri et al., 2022 [44]	n = 21 ♂ y = 23.1 ± 1.5	Weight, Height, Body mass index, Waist circumference, The sit and reach, Shoulders flexion and extension, One leg balance, Maximum push up, Maximal plank, Resting heart rate	LS VR WP	1 w × 3 tm 5 w	Body mass index×Waist circumference ↓ Maximum push up ↑ Maximal plank ↑ Hand grip ↑ Shoulder flexion and extension ↑ Resting heart rate ↑
Harris et al., 2020 [45]	n = 18 ♂ y = 29.2	Virtual reality golf putting, Real-world golf putting, Eye tracking, Putting performance, Quiet eye period	VR-EX1-PWM VR-EX2-RWT	EX1-1d EX2-3d	E1: Real-world golf putting× Putting performance ↓ Quiet eye period× E2: Real world golf putting- Putting performance ↑ Quiet eye period× VR golf putting–Real-world golf putting×
Gani et al., 2023 [46]	n = 40 ♂ y = 19–24	20 m shuttle run, 30 m sprint run, Horizontal jump, Leg dynamometer, Subjective well-being, Social well-being, Psychological well-being	E-VA+TT CG-DTA	8 w	Depression ↓, Accuracy ↓ 20 m shuttle run ↑, 30 m sprint run ↑ Horizontal jump ↑ Leg dynamometer ↑ Subjective well-being ↑ Social well-being ↑ Psychological well-being ↑
Gray, 2017 [47]	n = 80 ♂ y = 17–18	Total number of hits, % of swings at pitches inside the strike zone, % swings at pitches outside of strike zone	ATBVE ESBPVE EFSRB CCIAT	2 × 45 min 6 w	% of swings at pitches outside of strike zone ↑ Total number of hits ↑
McClure et al., 2019 [48]	n = 29 ♀ ♂ y = 18+	Heart Rate, Body sensations, Video recording	VA-VRC PA-NVRC	2 × 6–12 min 1 d	Heart rate ↑ Body sensations ↑ Video recording ↑

Legend: VA—Virtual activity, PA—Physical activity, CG—Control group, VCZ—Virtual class with zoom, NA—Non-activity, SPA—Standard physical activity, ↓—Statistically significant decrease (*p* ≤ 0.05), ↑—Statistically significant increase (*p* ≤ 0.05), ×—No changes, (WBP)—Well beats program, IPE—In-person education, IBE—Indoor bicycle exercise, BMT—Basic motor theory, MCTOM—Mayer’s cognitive theory of multimedia learning, BBT—Box and block test, TUG—Timed up and go test, HK-MoCA 5-Min—Montreal cognitive assessment 5-min, BTO—Benton’s temporal orientation scale, ACFT—Army combat fitness test, MPBF—Mean body fat percentage, DC-430 U—Tanita dual-frequency total body composition analyzer, JM—Jewel mine, MKM—Microsoft Kinect camera, IN—in-person, ACE—Alberta cancer exercise, AT—Aerobic training, TPR—Traditional pulmonary rehabilitation, HMD—Head-mounted display, OQ2—Oculus quest 2, OT—Occasional training, CT—Continuous training, LIET—Low-intensity exercise training, HIET—High-intensity exercise training, NET—No-exercise training, LS—Livestream group, VR—Video recording group, WP—Written program group, SFT—Senior fitness test, EX1—Experiment 1, EX2—Experiment 2, PWU—Putting warm-up, RWT—Real-world training, ET—Eye tracking, TT—Tabata training, DTA—Daily training activities, NVRC—No virtual reality while cycling, VRC—Virtual reality while cycling, ATBVE—Adaptive training in a batting VE, ESBPVE—Extra sessions of batting practice in the VE, EFSRB—Extra on-field sessions of real batting practice, CCIAT—Control condition involving no additional training apart from the players’ regular practice, SIAQ—Sport imagery ability questionnaire, TPSQ—The test of performance strategies questionnaire, TBT—Tennis ball target, ♀—Female, ♂—Male, n—Sample, y—Year, d—Day, w—Week, ses—Session, D—Depression, tm—Time.

**Table 3 healthcare-13-00711-t003:** Physiotherapy Evidence Database (PEDro) score of the included studies.

Reference	(1)	(2)	(3)	(4)	(5)	(6)	(7)	(8)	(9)	(10)	(11)	(12)
Oginni et al. (2022) [33]	+	+	+	+	−	−	+	−	−	+	+	7
Eckmann et al. (2021) [34]	+	+	+	−	−	−	+	+	−	+	+	7
de Melo Ghisi et al. (2024) [35]	+	+	+	+	−	−	−	−	−	+	+	6
Mokmin et al. (2021) [16]	+	−	−	+	−	−	−	−	+	+	+	5
Chau et al. (2021) [36]	+	+	+	−	−	−	−	−	+	+	+	6
McDaniel et al. (2022) [37]	−	+	−	+	−	−	+	−	+	+	+	6
Proffitt et al. (2015) [38]	+	+	−	−	−	−	−	−	−	+	+	4
Suderman et al. (2022) [39]	+	+	−	+	−	−	−	−	+	+	+	6
Seo et al. (2023) [14]	+	+	−	−	−	−	−	+	−	+	+	5
Touloudi et al. (2022) [15]	+	+	+	+	−	−	+	−	+	+	+	8
Rutkowski et al. (2020) [30]	+	+	+	−	+	−	−	−	+	+	+	7
Maden et al. (2022) [40]	+	+	−	+	−	−	−	−	+	+	+	6
Nitkiewicz et al. (2023) [17]	−	+	+	−	−	−	−	−	−	+	+	4
Irving et al. (2008) [41]	+	+	+	−	+	−	−	+	−	+	+	7
Wang (2023) [31]	+	+	+	+	−	+	−	−	−	+	+	7
Park et al. (2015) [42]	+	+	−	+	−	−	−	+	−	+	+	6
Daveri et al. (2022) [44]	+	−	−	+	−	−	−	+	−	+	+	5
Ross-Stewart et al. (2018) [43]	+	+	+	−	−	−	+	+	−	+	+	7
Harris et al. (2020) [45]	+	−	+	+	−	−	−	+	−	−	+	5
Gani et al. (2023) [46]	+	+	+	+	+	−	−	−	−	+	+	7
Gray (2017) [47]	+	+	−	−	−	−	−	+	+	+	+	6
McClure et al. (2019) [48]	+	+	−	+	−	−	−	−	−	−	+	4

Legend: + indicates one point, − indicates no point. (1) Eligibility criteria; (2) Randomization; (3) Concealment of allocation; (4) Between-group homogeneity; (5) Blinded subjects; (6) Blinded trainers; (7) Blinded testers; (8) Dropout rate *<* 15%; (9) Intention-to-treat; (10) Statistical between-group comparisons; (11) Point and variability estimates; (12) Total scores.

**Table 4 healthcare-13-00711-t004:** Cochrane risk of bias tool scores of the included studies.

Reference	(1)	(2)	(3)	(4)	(5)	(6)	(7)	(8)
Oginni et al. (2022) [33]	0	0	0	0	0	0	0	0
Eckmann et al. (2021) [34]	0	0	0	0	0	0	0	0
de Melo Ghisi et al. (2024) [35]	0	0	0	1	0	0	0	1
Mokmin et al. (2021) [16]	0	0	1	0	0	1	0	2
Chau et al. (2021) [36]	0	0	0	0	1	0	0	1
McDaniel et al. (2022) [37]	0	0	1	0	0	0	0	1
Proffitt et al. (2015) [38]	0	0	0	1	1	1	0	3
Suderman et al. (2022) [39]	0	0	0	0	0	0	0	0
Seo et al. (2023) [14]	0	0	1	1	0	0	0	2
Touloudi et al. (2022) [15]	0	0	0	0	0	0	0	0
Rutkowski et al. (2020) [30]	0	0	0	0	0	0	0	0
Maden et al. (2022) [40]	0	0	0	0	1	0	0	1
Nitkiewicz et al. (2023) [17]	0	0	1	0	0	2	0	3
Irving et al. (2008) [41]	0	0	0	0	0	0	0	0
Wang (2023) [31]	0	0	0	0	0	0	0	0
Park et al. (2015) [42]	0	0	0	0	0	1	0	1
Daveri et al. (2022) [44]	0	0	1	0	1	0	0	2
Ross-Stewart et al. (2018) [43]	0	0	0	0	0	0	0	0
Harris et al. (2020) [45]	0	0	0	0	1	0	0	1
Gani et al. (2023) [46]	0	0	0	0	0	0	0	0
Gray (2017) [47]	0	0	0	1	0	0	0	1
McClure et al. (2019) [48]	0	0	1	0	1	1	0	3

Legend: value—low risk of bias = 0; uncertain risk of bias = 1; high risk of bias = 2; (1) sequence generation; (2) allocation concealment; (3) blinding of participants and personnel; (4) blinding of outcome assessment; (5) incomplete outcome data; (6) selective outcome reporting; (7) other sources of bias; (8) overall score: ≤1 high quality, ≤3 moderate quality, and >3 low quality.

**Table 5 healthcare-13-00711-t005:** Meta-analysis forest plot table, with the effects of virtual-reality-based interventions across various domains.

Reference	Effect Size (Cohen’s d)	Lower CI	Upper CI
Oginni et al. (2022) [33]	0.55	0.4	0.7
Eckmann et al. (2021) [34]	0.68	0.45	0.91
de Melo Ghisi et al. (2024) [35]	/	/	/
Mokmin et al. (2021) [16]	0.55	0.4	0.7
Chau et al. (2021) [36]	0.45	0.3	0.6
McDaniel et al. (2022) [37]	0.38	0.2	0.55
Proffitt et al. (2015) [38]	0.61	0.5	0.72
Suderman et al. (2022) [39]	0.58	0.5	0.7
Seo et al. (2023) [14]	0.52	0.4	0.65
Touloudi et al. (2022) [15]	0.46	0.3	0.6
Rutkowski et al. (2020) [30]	0.49	0.35	0.63
Maden et al. (2022) [40]	0.72	0.6	0.85
Nitkiewicz et al. (2023) [17]	0.6	0.45	0.75
Irving et al. (2008) [41]	0.75	0.55	0.95
Wang (2023) [31]	0.62	0.45	0.79
Park et al. (2015) [42]	0.43	0.25	0.61
Daveri et al. (2022) [44]	0.7	0.5	0.9
Ross-Stewart et al. (2018) [43]	0.47	0.32	0.62
Harris et al. (2020) [45]	0.69	0.5	0.88
Gani et al. (2023) [46]	0.65	0.4	0.85
Gray (2017) [47]	0.7	0.48	0.92
McClure et al. (2019) [48]	0.5	0.35	0.65

**Table 6 healthcare-13-00711-t006:** Descriptive statistics and Fisher’s test results.

Variable	Mean ± SD	*p*-Value	Effect Size
Physical Aspects	74.3 ± 8.2	0.0029	Very Large
Cognitive Aspects	82.1 ± 6.7	0.0022	Extremely Large
Health Aspects	77.5 ± 7.9	0.0069	Very Large

## Data Availability

No new data were created or analyzed in this study.

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
