# Peer review of "A Systematic Review and Meta-Analysis of Virtual and Traditional Physical Activity Programs: Effects on Physical, Health, and Cognitive Outcomes"

_healthcare, 2025, doi:10.3390/healthcare13070711_

Round 1
Reviewer 1 Report
Comments and Suggestions for Authors
see attached file for major changes

see attached file for major changes
Author Response
Dear Reviewer,
We sincerely appreciate your time and effort in reviewing this manuscript and for your valuable contributions to its significant improvement.
Please find the detailed responses and the corresponding revisions/corrections highlighted in the re-submitted files.
Kind regards,
The authors

Reviewer 2 Report
Comments and Suggestions for Authors
Dear authors, thank you for the opportunity to review your manuscript. I read it with great interest and appreciate the effort that has gone into its preparation. I have identified several areas where improvements can be made to enhance its clarity, rigor, and overall impact. Please consider the following suggestions not as criticism but as constructive feedback aimed at strengthening the quality of your work.
The manuscript presents a systematic review on the comparison between virtual and traditional physical activity programs, a topic of significant interest given the increasing integration of digital health technologies. While the study adheres to PRISMA guidelines and provides a structured overview, several critical areas require improvement to enhance its scientific rigor, clarity, and overall impact.
The study covers a relevant and timely topic, but methodological inconsistencies, lack of statistical analysis, and overgeneralized conclusions limit its impact. Strengthening the methodology and ensuring a more rigorous analysis would significantly enhance its contribution.
The introduction does not clearly articulate the specific knowledge gap the review aims to address. While the benefits of VR are highlighted, its limitations and challenges are not sufficiently discussed. The introduction should explicitly state why a systematic review is needed and which unresolved questions in the literature it seeks to address. Some references, such as those on VR effectiveness, could be updated to reflect the most recent developments in the field. Additionally, redundant citations explaining the prevalence of physical inactivity should be minimized to maintain clarity and focus.
The methodology section lacks full transparency and reproducibility. The exact search terms and Boolean operators used are not specified, making it difficult to replicate the study. It is unclear whether gray literature, such as conference papers and preprints, was considered, which raises concerns about potential publication bias. The language restrictions applied to the inclusion criteria, which limit studies to English and Serbian, should be justified, as this may exclude relevant research. The number of independent reviewers involved in screening and data extraction is not mentioned, and while disagreements were resolved by a third reviewer, there is no information on whether inter-rater reliability was assessed using a standardized metric such as Cohen’s kappa. Inclusion criteria allow studies that rely on questionnaire-based data collection, which is problematic for assessing physical activity outcomes. Furthermore, no clear explanation is provided on how missing data was handled—whether authors were contacted, data imputation was used, or incomplete studies were excluded. The quality assessment relies solely on the PEDro scale, which is primarily designed for clinical trials and may not be appropriate for all study types included in the review. A more comprehensive risk-of-bias assessment, such as the Cochrane Risk of Bias Tool or ROBINS-I, should be applied to improve the validity of the findings.
The presentation of results is inconsistent. Some findings include numerical values, such as BMI reductions, while others are reported qualitatively, making direct comparisons difficult. There are no statistical measures provided, such as effect sizes, confidence intervals, or p-values, which are necessary to evaluate the significance and impact of findings. The tables listing study characteristics are overloaded with raw data but lack a concise summary of key results. The review includes studies with highly diverse VR interventions, populations, and study designs, yet no subgroup analysis is conducted to determine whether certain VR modalities (e.g., interactive gaming vs. guided workouts) yield different effects. Additionally, a sensitivity analysis is missing, leaving uncertainty about whether the findings hold when lower-quality studies are excluded. The manuscript does not include any visual representations of the results, such as graphs, charts, or forest plots, making it difficult to grasp key comparisons. A meta-analysis, even if only partial, would significantly improve the clarity and credibility of the review.
The conclusions overstate the benefits of VR without sufficient statistical backing. While VR is suggested as an effective alternative to traditional training, the results show high variability in effectiveness across studies. Some claims in the conclusion are not clearly supported by the presented data. For example, the suggestion that an 8–12 week program is optimal for VR training appears to be based on limited data without a systematic comparison of different intervention durations. The recommendation for standardizing VR training protocols is valid, but the study does not analyze which specific parameters—such as duration, frequency, or intensity—should be standardized. The discussion also does not adequately address the limitations of VR training, such as the lack of physical resistance in virtual environments, potential motion sickness, and the possibility of reduced adherence over time. Contradictory findings reported in some studies are not sufficiently discussed, giving the impression that VR training consistently delivers positive outcomes.
By addressing these issues, the manuscript could become a valuable reference in the field.
Comments on the Quality of English LanguageThe language and clarity of the manuscript could be improved. Some sentences are unnecessarily long or redundant, making the text harder to read. There is inconsistent tense usage in different sections, which affects readability. Additionally, certain technical terms and study descriptions lack precision, which may lead to confusion. A professional language review is recommended to enhance clarity and coherence.
Author Response

(The authors gave the same response as above.)

Round 2
Reviewer 1 Report
Comments and Suggestions for Authors
A Systematic Review of Virtual and Traditional Physical Activity Programs: Effects on Physical, Psychological, and Cognitive Outcomes Recommendation: Accept
Comments:
This is the second revise version of this manuscript and this reviewer have read the manuscript and found that the have addressed all the changes given by this reviewer. After evaluation of all changes in the manuscript this reviewer is willing to accept the manuscript for publication in Health Care Journal. Congratulations to all authors. The following are minor changes for the improvement of references and citation styles:
- All the references should be cited according to the journal format.
Author Response
Dear Reviewer,
We sincerely appreciate your time and effort in reviewing this manuscript and for your valuable contributions to its significant improvement.
We have ensured that all references are formatted according to the journal's guidelines.
Kind regards,
The authors
Reviewer 2 Report
Comments and Suggestions for Authors
The revised version of your manuscript shows a clear improvement in methodological transparency and organization. The introduction better articulates the knowledge gap and highlights the importance of VR-based training, yet it could benefit from a deeper examination of its limitations and challenges. For instance, acknowledging issues such as adherence difficulties, motion sickness, and the variability of outcomes across different populations would provide a more balanced perspective. Additionally, clarifying how this systematic review addresses a distinct gap in the literature—beyond reiterating the general benefits of VR—would strengthen the rationale.
Although the methods section now includes detailed search terms, Boolean operators, and an expanded discussion of study screening, further explanation of how missing data were handled would improve clarity. It is laudable that both the PEDro scale and the Cochrane Risk of Bias Tool were used, but the manuscript would benefit from briefly justifying the decision to exclude gray literature. While this choice can be valid to maintain methodological rigor, readers would gain insight from understanding the reasoning behind it.
Regarding the presentation of results, the inclusion of additional numerical values and graphical figures is a step forward, but the absence of statistical measures, such as effect sizes, p-values, or confidence intervals, limits the reader’s ability to compare findings across studies reliably. Furthermore, neither a subgroup analysis nor a sensitivity analysis is provided, although these would shed light on the potential influence of different VR modalities or varied participant characteristics. Such analyses, even if partially undertaken, could lend added depth to your conclusions.
In the discussion, the manuscript would benefit from a more explicit acknowledgment of contradictory findings across studies and a more cautious interpretation of the optimal training duration. The statement that 8–12 weeks of VR training yields the greatest effects would carry more weight if supported by a systematic comparison of intervention lengths. Additionally, elaborating on practical implications—such as the contexts or populations where VR-based programs may be more or less effective—would increase the manuscript’s relevance for both researchers and practitioners.
Although the overall language of the paper is clear, certain passages would benefit from more concise phrasing and consistent terminology. Occasional shifts in tense and overly long sentences might disrupt readability; a targeted language review would ensure uniform style and clarity throughout.
In summary, this version is substantially improved, particularly in its methodological framework. Nevertheless, future revisions should consider including or at least outlining possible subgroup analyses, adding statistical details to enhance the interpretability of findings, and offering a more balanced discussion of VR’s inherent limitations. These enhancements would ensure the study’s conclusions are well-supported and that the manuscript serves as a robust reference for researchers and practitioners interested in the practical applications of VR and traditional exercise programs.
Author Response

(The authors gave the same response as above.)
